# Ecological Risk Due to Heavy Metal Contamination in Sediment and Water of Natural Wetlands with Tourist Influence in the Central Region of Peru

**María Custodio** [1,*] , **Anthony Fow** [2] , **Fernán Chanamé** [1] , **Edith Orellana-Mendoza** [1] , **Richard Peñaloza** [1] , **Juan C. Alvarado** [3] , **Deyvis Cano** [1] **and Samuel Pizarro** [1]

[1] Centro de Investigación de Medicina en Altura y Medio Ambiente, Universidad Nacional del Centro del Perú, Av. Mariscal Castilla N° 3909, Huancayo 12006, Peru; fchaname@uncp.edu.pe (F.C.); eporellana@uncp.edu.pe (E.O.-M.); drach_89@hotmail.com (R.P.); deyviscano02@gmail.com (D.C.); sam200592@hotmail.com (S.P.)

[2] Facultad de Ingeniería Ambiental y de Recursos Naturales, Universidad Nacional del Callao, Av. Juan Pablo II 306, Bellavista-Callao 07011, Peru; ajfe081@gmail.com

[3] Universidad Nacional Intercultural "Fabiola Salazar Leguía" de Bagua, Jr. Comercio N 128, Bagua 01720, Peru; jalvarado@unibagua.edu.pe

* Correspondence: mcustodio@uncp.edu.pe

**Abstract:** In this study, the quality of sediment and surface water in two natural wetlands, Paca and Tragadero, in the central region of Peru was evaluated using pollution indices, including the geoaccumulation index, pollutant load index, modified pollution degree, potential ecological risk index, and site rank index, for four heavy metals. Principal component analysis was used to identify potential metal contaminant sources. The determination of Fe, Zn, Pb, and As was performed by flame atomic absorption spectrophotometry. The average concentrations of metals in the sediments of both lagoons decreased in the order Fe > Zn > Pb > As. The analysis of the contamination indices determined that As and Pb are the elements that contribute the most to environmental degradation in both wetlands. There is a strong correlation between the values of potential ecological risk and the modified degree of contamination, revealing that the Paca wetland has a moderate degree of contamination and potential ecological risk, while Tragadero presents a high degree of contamination and considerable potential ecological risk. The application of the site rank index showed that more than 50% of the sampling sites have between high and severe contamination. The principal component analysis presented 79.2% of the total variance. Finally, the results of this study are essential in order to carry out preventive actions for environmental protection in these lake ecosystems of great importance for many activities, such as bird watching.

**Keywords:** ecological risk; site rank index; contamination; heavy metal; natural wetlands; lagoon

## 1. Introduction

Contamination of aquatic ecosystems by toxic metals is a worldwide concern due to their toxicity, non-biodegradable nature, persistence, and accumulation in various habitats [1,2]. Although heavy metals have a low solubility, once released into the aquatic environment through various sources [3,4], they are deposited in bottom sediments. The quality of aquatic ecosystems is greatly influenced by intensive urbanization, population growth, urban runoff pollutants, domestic and industrial wastewater effluents, and agriculture [5]. The release of heavy metals from sediments into the water column under favorable conditions makes the aquatic environment extremely vulnerable to contamination [1,6,7].

Heavy metals are readily absorbed by organisms, and can bioaccumulate and enter the food chain [8,9] and pose risk to human health and ecosystem integrity [10,11]. In humans, heavy metals can alter metabolic processes, leading to their accumulation in the liver and kidneys and inducing systemic toxicity [12]. Many studies around the world report that

lead (Pb) concentrates mainly in bones, teeth, and fatty tissue, leading to the depletion of essential nutrients and immune defenses [13]. Arsenic (As) in excessive levels can cause cancer, dermal, respiratory, cardiovascular, gastrointestinal, hematological, hepatic, renal, neurological, developmental, reproductive, and immune problems [14]. However, some heavy metals such as iron (Fe) and zinc (Zn) are micronutrients and enzyme cofactors essential for the normal development of biological processes, and only pose a threat to health when they exceed permissible limits.

Assessment of the state of aquatic ecosystems is of high priority to determine the level of heavy metal contamination [15]. Contamination indices have been widely used to assess the contamination of surface water [16,17], fish [18,19], and sediment [20–22]. In the assessment of the sediment contamination level, individual and complex indices are usually applied [23]. Individual indices evaluate the individual impact of each metal, such as the contamination factor, enrichment factor, and geoaccumulation index [24]. Complex indices evaluate the impact of metals jointly, such as the pollution load index, modified pollution degree, and potential ecological risk [25,26].

The natural wetlands of Paca and Tragadero are aquatic ecosystems of ecological and economic importance. They are habitats for different species of birds that are tourist attractions. They are also an important source of water resources for the surrounding human populations and agricultural and livestock activities in the area. To our knowledge, to date, no comprehensive and exhaustive research has been conducted on the contamination levels and ecological risks of heavy metals in the sediments of the Paca and Tragadero wetlands. In this sense, the present investigation focused on evaluating the ecological risk of toxic metal contamination in the sediments of natural wetlands in the central region of Peru, applying individual and complex indices, determining their relationship with the concentration of heavy metals in water, and identifying the sources of contamination through a combination of multivariate statistical analyses.

## 2. Materials and Methods

### 2.1. Study Area

The Paca and Tragadero wetlands have important bodies of water with the same name as the wetlands. They are located in the Junín region, in the Central Andes of Peru, in the northeastern Mantaro Valley at $11°46'48''$ S and $75°30'13''$ N. Paca Lake is fed by surface runoff from other lakes and springs, and is located at 3365 masl. Tragadero Lagoon is located in the Yanamarca River micro-basin at 3460 masl. The aquatic systems of both lagoons cover an area with submerged and emergent macrophytes dominated by *Scirpus californicus* "totora", which reach a height of 3 to 3.5 m and in areas with little flooding between 0.4 and 0.5 m. The region's climate is cold, with an average annual temperature of 11.4 °C and annual precipitation of 649 mm, with rainy periods from January to March and dry periods from June to August. Paca Lagoon is mainly used for tourism and agriculture. In both wetlands, the richness of birds is important, but it is more significant in Tragadero where bird watching is an attraction. Drainage from restaurants, the massive use of detergents on the shore, waste left by visitors, and poor management of the surrounding micro-watershed are the main problems in Paca Lagoon. Agricultural runoff, water extraction, and wastewater from Chocón's population centers affect Tragadero Lagoon. Both lagoons receive tributary rivers that flow through other population centers (Figure 1).

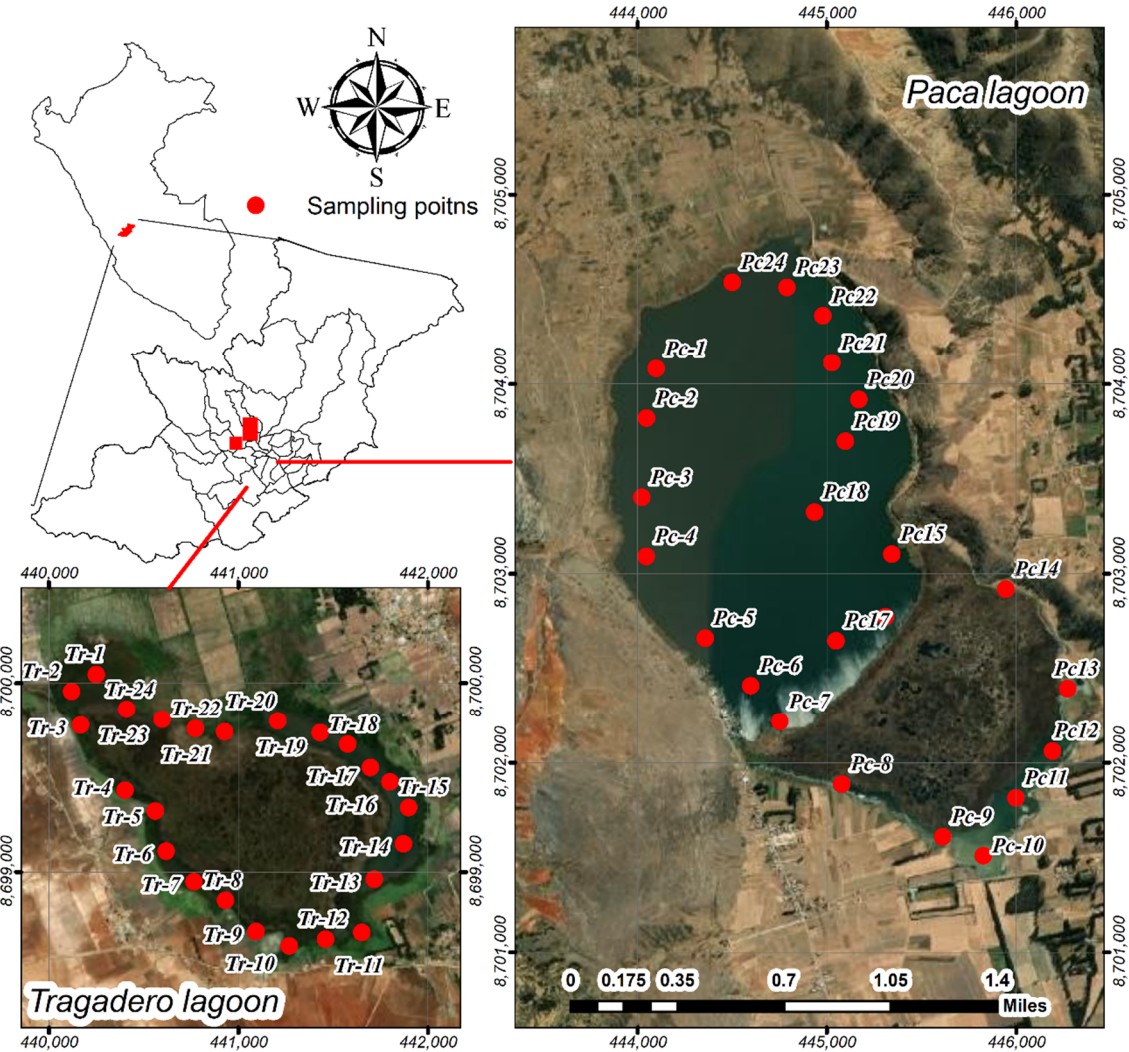

**Figure 1.** Location of water and sediment sampling sites in the Paca and Tragadero lagoons.

## 2.2. Sampling and Analytical Procedure

Six sampling stations were established at each lagoon with four sites per station. A total of 48 water samples were collected (24 samples at Paca Lagoon and 24 samples at Tragadero Lagoon) during 2018. Water samples were collected in 500 mL plastic bottles, previously sterilized and rinsed with distilled water. Each sample was preserved by adding 1.5 mL of concentrated nitric acid to one liter of water [27]. After collection, the water samples were transported to the laboratory and stored at 4 °C for preparation and analysis. Surface sediment samples (10 cm) were collected at the same sites as the water samples using a cylindrical stainless steel dredge. Composite samples were formed at each sampling site and were packed in sterile, hermetically sealed polypropylene bags. The water and sediment samples were conditioned separately in a cooler under refrigerated conditions (4 °C) and were sent to the laboratory for analysis.

The operational methods used for the determination of heavy metals corresponded to those specified in the international analytical standards of the American Public Health Association [27] and the Food and Agriculture Organization of the United Nations [28]. Water samples were filtered through 0.45 μm membrane filters. Digestion of the samples was performed from 250 mL of water, brought to boiling until 100 mL was obtained. Then, 5 mL of nitric acid and 5 mL of concentrated hydrochloric acid were added for the destruction of the organic matter, and again it was brought to boiling (until the water was consumed and a pasty consistency residue was obtained). It was allowed to cool and then

10 mL of distilled water was added, filtered, and gauged in a volumetric flask of 100 mL, with 1% nitric acid.

The sediment samples were dried for five days in rectangular containers exposed to solar energy to evaporate all the water, leaving the sediment completely dry. The sediment was then dewatered and sieved through a 2 mm stainless steel mesh sieve to remove the stones and plant debris. The sieved sediment was placed in an electric oven at 60 °C for 24 h and the completely dry samples were pulverized in a mill. First, 1 g of each sediment sample was weighed into a 100 mL beaker and 10 mL of nitric acid was added and allowed to act for a few seconds for disintegration of the organic matter. Then, 10 mL of hydrochloric acid was added and left to act for one minute to dissolve the salts. It was then taken to boiling for five minutes until the sample achieved a pasty consistency, it was removed from the stove and again 10 mL of hydrochloric acid was added to dissolve the remains of our adhered in the walls of the glass. Subsequently, the sample was transferred to a 100 mL beaker for its homogenization and gauging with distilled water to 100 mL, and then it was filtered. The concentration of Pb, Fe, and Zn in water (mg $L^{-1}$) and sediment (mg $Kg^{-1}$) was determined by graphite furnace atomic absorption spectrophotometry (GF-AAS) and As by hydride generation atomic absorption spectrophotometry (HGAAS), using an AA-6800 Atomic Absorption Spectrophotometer, Varian AA240.

### 2.3. Quality Control

Analytical grade reagents and standard stock solutions were supplied by Merck (Darmstadt, Germany). Double deionized water (Milli-Q System, Millipore, Darmstadt, Germany) was used for the preparation of all of the solutions. Element standard solutions used for calibration were prepared by diluting stock solutions of 1000 mg $L^{-1}$ of Cu, Pb, Zn, Fe, and As. Working standards of 0.001, 0.01, 0.1, 0.1, 1.0, and 2.0 mg $L^{-1}$ were prepared from an average standard solution of 100 mg $L^{-1}$ with 1% nitric acid. All of the glassware used were cleaned by immersion in dilute nitric acid for 24 h. They were then rinsed with deionized water before use. The precision of the analytical procedures expressed as the relative standard deviation was less than 5% for all metals analyzed by atomic absorption spectrophotometry. All of the analyses were carried out in triplicate and the results were expressed as the mean.

### 2.4. Assessment of Heavy Metal Contamination of Sediments

The evaluation of the heavy metal contamination of sediments in the Paca and Tragadero wetlands was carried out based on individual and complex indices. The individual indices used included the contamination factor (Cf), enrichment factor (EF), and geoaccumulation index ($I_{geo}$). The complex indices included pollution load index (PLI), modified pollution degree (mCd), and potential ecological risk (Ri). These indices are widely used for the evaluation of toxic metals in sediment samples around the world [23,26].

#### 2.4.1. Determination of Background Values

A determining factor for the correct application of indices in the assessment of sediment contamination is the designation of the background levels to be used, as an incorrect choice can lead to the recognition or non-recognition of anomalous values present in the sediments of the ecosystem under study [29]. The appropriate choice of background values plays an important role, and should be based on specific local background criteria or otherwise widely applicable reference values [26]. Among the most commonly used background levels are those of the upper continental crust [30]. In Peru, there are no studies that evaluate the background levels of metals in soil [31]. Similar studies have used the values of the upper continental crust, obtaining consistent results in accordance with the reality being studied [32–34]. This study also uses the background levels proposed by Taylor and Mclennan [30].

### 2.4.2. Individual and Complex Indices to Assess Heavy Metal Contamination of Sediments
Contamination Factor (Cf)

The Cf is the ratio between the concentration of each metal in the sediment and the background value. It is applied to quantify the metal contamination status of the sediment [23,35]. Cf is calculated using Equation (1).

$$\text{Cf} = \frac{C_m \text{ sample}}{C_m \text{ background}} \tag{1}$$

where $C_m$ sample is the concentration of metal in the sediment and $C_m$ background is the concentration of metal in upper continental crust (Table S1).

Enrichment Factor (EF)

EF is used to estimate the anthropogenic impact of heavy metals in sediments [29]. This method typically uses iron or aluminum as a tracer to distinguish natural from anthropogenic sources [23,36]. EF is calculated using Equation (2)

$$\text{EF} = \frac{(C_m \text{ sample} \times C_{Fe} \text{ background})}{(C_m \text{ background} \times C_{Fe} \text{ sample})} \tag{2}$$

where $C_m$ sample is the concentration of metal in the sediment, $C_m$ background is the concentration of metal in upper continental crust, $C_{Fe}$ sample is the concentration of iron in the sediment, and $C_{Fe}$ background is the concentration of iron in the upper continental crust (Table S1).

Geoaccumulation Index ($I_{geo}$)

$I_{geo}$ is used to establish the difference in heavy metal concentrations between samples and background values that exist naturally in the continental crust [37]. $I_{geo}$ is calculated using Equation (3).

$$I_{geo} = \log_2 \frac{C_n}{1.5 \times B_n} \tag{3}$$

where $C_n$ is the concentration of metal in the sediments, $B_n$ is the concentration of metal in the upper continental crust, and 1.5 is a factor applied to minimize the effect of possible variations in the background values (Table S1).

Pollution Load Index (PLI)

The pollution load index allows (PLI) for knowing the degree of contamination of sediments by heavy metals and is used to determine the degree of deterioration [38,39]. PLI is calculated using Equation (4).

$$\text{PLI} = (\text{CF1X CF2X CF3X} \ldots \text{CFn})^{1/n} \tag{4}$$

where $\text{CF}_n$ is the contamination factor of each metal and *n* is the number of metals evaluated. Furthermore, the categories of PLI are given in Table S2.

Modified Degree of Contamination (mCd)

mCd allows for assessing the overall degree of sediment contamination by heavy metals [29], and for incorporating as many metals as the study includes [26]. mCd is calculated by Equation (5).

$$\text{mCd} = \frac{\sum_{i=1}^{i=n} C_f^i}{n} \tag{5}$$

where $C_f$ is the contamination factor, *i* represents the "*i*th" metal, and *n* is the number of metals evaluated (Table S2).

Potential Ecological Risk (Ri)

The potential ecological risk (Ri) is an index applicable for the evaluation of the degree of ecological risk caused by heavy metal concentrations in water, air, and soil [35]. Ri is calculated by Equation (6).

$$E_r^i = T_r^i \times CF_r^i \; ; \; \text{Ri} = \sum_{i=1}^{n} E_r^i \tag{6}$$

where $E_r$ is the ecological risk index of each metal, $T_r$ is the biological toxic factor of each metal, and $i$ represents the "$i$th" metal (Table S2).

Site Rank Index (SRI)

SRI allows for a better understanding of the sediment pollution status with respect to the environment under study, as it compares the pollution level of the sampling sites with respect to the concentrations of elements analyzed under same metrics, avoiding the comparison in different classifications of each pollution index (CF, $I_{geo}$, and EF) [21,36]. The *SRI* is calculated using Equation (7).

$$W = \frac{\sum n_i}{\sum i} \; ; \; SRI = \frac{W}{S} \times 100 \tag{7}$$

where $S$ is the number of sampling stations, $n$ is the pollution rank of the site in ascending order (to each value used: $I_{geo}$, RI, EF, etc.), and $i$ represents the "$i$th" metal (Table S2).

*2.5. Evaluation of Heavy Metal Contamination of Water*

To evaluate the contamination levels in the water samples, the results were compared with the Peruvian regulations in force through the environmental quality standards for water (aquatic environment conservation category), and the site classification index (SRI) was calculated to standardize the contamination levels of the samples, in order to compare with the results for the sediment samples.

*2.6. Statistical Analysis*

Statistical analysis and data processing were performed using InfoStat (Version 2020) and R (Version 4.0.5) software. The modified Shapiro−Wilks test was used to determine the normality of the data. Pearson correlation analysis was used to identify the degree of correlation between toxic metals, and linear regression was used to measure the degree of association between the degree of modified contamination (mCd) and potential ecological risk (Ri). Principal component analysis (PCA) was used to identify possible sources of toxic metal contamination in the sediments of both lagoons.

## 3. Results and Discussion

*3.1. Concentration and Distribution of Heavy Metals in Sediment and Water*

Descriptive statistics for heavy metals in the sediment and water from the Paca and Tragadero lagoons, probable effect concentration (PEC) threshold values [40], upper continental crust (UCC) reference values, Canadian Interim Sediment Quality Guidelines of the Canadian Council of Ministers of the Environment (ISQG-CCME) values (ISQG-CCME) [41], and the Peruvian environmental quality standards (EQS) for water from lagoons and lakes in the aquatic environment conservation category of the Ministry of Environment [42] are presented in Table 1.

**Table 1.** Comparison of the mean concentration of heavy metals in the sediment and water of the Paca and Tragadero lagoons in this study with threshold values, background values, quality guidelines, and mean concentrations of other lagoons.

| Lagoons | | Sediment (mg kg⁻¹) | | | | | | | | | | | |
|---|---|---|---|---|---|---|---|---|---|---|---|---|---|
| | | Pb | | | Zn | | | Fe | | | As | | |
| Paca | Mean ± SD | 45.80 | ± | 4.11 | 85.16 | ± | 11.95 | 9530.76 | ± | 764.27 | 13.67 | ± | 1.98 |
| | Rank | 53.96 | – | 35.56 | 105.52 | – | 67.62 | 10,875.00 | – | 7784.20 | 17.53 | – | 10.35 |
| Tragadero | Mean ± SD | 49.71 | ± | 5.53 | 71.59 | ± | 6.71 | 17,170.48 | ± | 4340.40 | 22.99 | ± | 3.64 |
| | Rank | 59.52 | – | 41.52 | 84.64 | – | 62.62 | 21,649.40 | – | 10,659.20 | 28.83 | – | 16.21 |
| | CEP | 128.0 | | | 459.0 | | | NA | | | 33.0 | | |
| | UCC | 20.0 | | | 71.0 | | | 35000.0 | | | 1.50 | | |
| | ISQG-CCME | 30.20 | | | 124.0 | | | NA | | | 7.24 | | |
| Junín Lake [43] | | 24.06 | | | 76.18 | | | 319.76 | | | 17.50 | | |
| Titicaca Lake [44] | | 74.97 | | | 323.24 | | | 433.38 | | | NA | | |
| Qaroun Lake [45] | | 10.0 | | | 130.0 | | | 174,100.0 | | | NA | | |
| Lihu Lake [46] | | 74.46 | | | 102.21 | | | NA | | | 12.36 | | |
| Huixian wetland lake [47] | | 51.37 | | | 107.70 | | | NA | | | 14.0 | | |
| | | Water (mg L⁻¹) | | | | | | | | | | | |
| Paca | Mean ± SD | 0.0117 | ± | 0.0022 | 0.0800 | ± | 0.0044 | 0.0217 | ± | 0.0021 | 0.0042 | ± | 0.0009 |
| | Rank | 0.0160 | – | 0.0065 | 0.0855 | – | 0.0738 | 0.0235 | – | 0.0179 | 0.0057 | – | 0.0025 |
| Tragadero | Mean ± SD | 0.0189 | ± | 0.0047 | 0.0793 | ± | 0.0083 | 0.0388 | ± | 0.0045 | 0.0219 | ± | 0.0021 |
| | Rank | 0.0255 | – | 0.0090 | 0.0920 | – | 0.0642 | 0.0458 | – | 0.0282 | 0.0251 | – | 0.0184 |
| | EQS MINEN | 0.0025 | | | 0.12 | | | NA | | | 0.15 | | |
| | Standards CCME | 0.01 | | | 5.00 | | | NA | | | NA | | |
| Junín Lake [48] | | 0.003 | | | 0.028 | | | 0.073 | | | 0.005 | | |
| Titicaca Lake [44] | | 0.032 | | | 0.047 | | | 0.041 | | | NA | | |
| Strzelin Quarry Lakes [49] | | 0.003 | | | 0.021 | | | NA | | | NA | | |
| Manchar Lake [50] | | 0.025 | | | 0.024 | | | 0.197 | | | 0.018 | | |

NA: Not applicable or available.

The mean concentration of Pb, Zn, and As in the sediments of the Paca and Tragadero lagoons did not exceed the threshold values of the probable effect concentration. However, the mean concentrations of Pb (45.80 ± 4.11 and 49.71 ± 5.53 mg kg⁻¹, respectively) and As (13.67 ± 1.98 and 22.99 ± 3.64 mg kg⁻¹, respectively) in both lagoons greatly exceeded the upper continental crustal reference background values (Pb, 128 mg kg⁻¹; As, 33 mg kg⁻¹) and the Canadian Council of Ministers of the Environment's interim sediment quality guidelines values (Pb, 30.2 mg kg⁻¹; As, 7.24 mg kg⁻¹). The decreasing order of metal concentration in the sediments of the studied ponds was Fe > Zn > Pb > As. The distribution of heavy metal content in the sediments varied widely. The highest concentration of Pb (59.52 mg kg⁻¹), Fe (21,649.40 mg kg⁻¹), and As (28.83 mg kg⁻¹) was recorded in the Tragadero lagoon, and that of Zn (105.52 mg kg⁻¹) in the Paca lagoon (Tables S3 and S4). In comparison with other studies, the Pb, Zn, Fe, and As contents in the sediments of the Paca and Tragadero lagoons were relatively higher than those recorded in the lake of the Junín National Reserve [43]. This difference would be determined by multiple stressors, such as urbanization in both lakes and the strong tourist activity (restaurants and recreation) in Paca lagoon. In addition, the lake does not have an environmental management plan, as the lake in Junín National Reserve does. In contrast, the present results are lower than those reported in similar environments in Peru [44] and other regions of the world [45–47].

The mean concentration of Zn and As in the water from the Paca and Tragadero lagoons did not exceed Peru's environmental quality standard for the conservation of the aquatic environment (Zn, 0.12 mg L⁻¹; As, 0.15 mg L⁻¹) or the Canadian Council of Ministers of the Environment's standards for aquatic life (CCME) (Zn, 5.0 mg L⁻¹). The mean Pb concentration (Paca, 0.0117 ± 0.0022 mg L⁻¹; Tragadero, 0.0189 ± 0.0047 mg L⁻¹), exceeded the environmental quality standards for water (0.0025 mg L⁻¹) established by the national norm and the CCME aquatic life standards (0.01 mg L⁻¹). The 75% content of heavy metals in this study is higher than the metal contents recorded in Lake Junín [48], but slightly lower than the metal content reported for Lake Titicaca [44]. In comparison with the heavy metal contents of other lakes in the world [49], for Manchar Lake [50], the results obtained were similar.

The high concentrations of Pb, Zn, and As in the water bodies of the wetlands studied are due to wastewater discharges from human settlements located around the lagoons, tributary rivers that receive wastewater from the population centers they flow through, and runoff from agricultural areas adjacent to the wetlands. Increased heavy metal content could affect sediment geochemistry and cause deleterious effects on the biota, as metal retention suggests a high affinity for the sediment through adsorption reactions and the low geochemical mobility in water [51]. Therefore, the application of individual and complex indices and multivariate statistics allows for the identification of the level of contamination of the two lagoons [52].

### 3.2. Evaluation of Heavy Metal Contamination of Sediments by Individual and Complex Indices

Figure 2 shows the results of the individual heavy metal sediment contamination indices for the Paca and Tragadero lagoons. The EF values for Pb and As in both lagoons indicated moderately severe enrichment (5–10) and very severe enrichment (25–50), respectively, which could be attributed to a high anthropogenic impact on these two aquatic ecosystems. Of the sampling sites, 58% presented EF values for Pb of a moderate type (3–5) and 42% of moderately severe type (5–10). This Pb enrichment could be attributed to activities related to urban transport traffic [53], especially in the Paca lagoon, because of its proximity to the city. Variation in EF values for As was observed, as follows, 25% of the sampling sites in the Tragadero lagoon had severe enrichment (10–25), 58% had very severe enrichment (25–50), and 17% had extremely severe enrichment (>50). Meanwhile, in the Paca lagoon, 96% of the sampling sites showed very severe enrichment (25–50). These results reveal the enrichment of sediments with heavy metals from point sources [54] such as wastewater discharge from tourism, urban transport, and agricultural and livestock activities. The EF values for Zn were less than 1, which would indicate an absence of enrichment for this chemical element. Sediment samples in the Paca lagoon showed moderately severe enrichment for Pb, and very severe enrichment for As, while the Tragadero lagoon presented an enrichment that ranged from moderate to moderately severe for Pb, and a severe to extremely severe enrichment for As. The results of this study are supported by [55], which shows that high EF values can be attributed to industrial effluents, agricultural activities, domestic solid waste, and municipal wastewater.

Cf quantifies the contamination status for each metal in the sediment. The results found in the sediment samples indicate that the contamination factors for Pb, Zn, Fe, and As in the Paca and Tragadero lagoons are classified as low, moderate, and very high. The total sediment samples from the Tragadero and Paca lagoons presented Cf values for As greater than 6, classifying the lagoons with a very high level of contamination. The Cf values for Pb in the two ponds showed a level of contamination ranging from moderate (1–3) to considerable (3–6). The Cf values for Zn in the two ponds indicated a low (<1) to moderate (1–3) level of contamination. Of the samples, 87% and 46% of the samples showed a moderate level of contamination in the Paca and Tragadero lagoons, respectively. The Cf values for Fe in both lagoons presented values less than 1 in all of the sediment samples, which would indicate a low level of contamination for this element. The results obtained for Cf show that the sediments of the Paca and Tragadero lagoons present conditions of very high As contamination, moderate Pb contamination, and low and moderate Zn contamination, respectively. Human activities such as urbanization, discharge of domestic waste effluents, and urban transportation increased the level of contaminants in the wetlands studied.

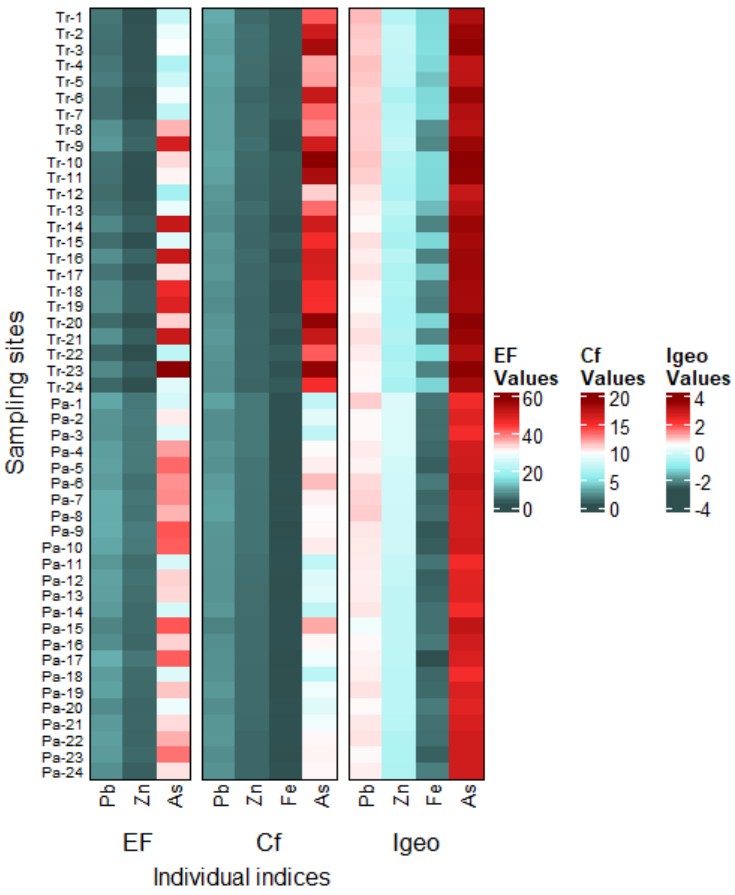

**Figure 2.** Contamination factor (Cf), enrichment factor (EF), and geoaccumulation index ($I_{geo}$) of heavy metals in the Paca (Pa) and Tragadero (Tr) lagoons.

Sediment $I_{geo}$ values for Pb indicate that the total sampling sites are slightly contaminated ($0 < I_{geo} < 1$) in both lagoons. However, the $I_{geo}$ values for As in the Paca and Tragadero lagoons indicate that 100% and 87.5% of the studied samples presented a moderate to heavily contaminated ($2 < I_{geo} < 3$) and heavily contaminated ($3 < I_{geo} < 4$) contamination level, respectively. The $I_{geo}$ results for Fe and Zn show that the two ponds are not contaminated ($I_{geo} \leq 0$). The $I_{geo}$ results show that both ponds are slightly contaminated by Pb and moderately to heavily contaminated by As. $I_{geo}$ values greater than 1.0 indicate an elevated heavy metal content and sediment contamination [56].

The integral analysis of the three individual indices used revealed that both lagoons are contaminated by As and Pb. The different classifications for each contamination index studied present different levels of contamination, from "light" or "low" to "heavily contaminated" or "extremely severe". Because of the wide range of classifications generated by the results of these individual indices, it is difficult to analyze the contamination levels of each sample. For this reason, and in order to eliminate arbitrary and different classifications of each individual index, SRI was used [36]. SRI allows for an objective comparison of the contamination indices such as Cf, $I_{geo}$, and EF [21].

Figure 3 shows the distribution of the individual indices and metal concentrations in the surface water samples according to the application of SRI. In the Paca and Tragadero lagoons, it can be observed that the SRI of each of the indices applied to sediments and the SRI of surface water present a similar distribution according to the values found. In the sampling sites of both lagoons, 50% present conditions of high and severe contamination due to the presence of the metals evaluated. According to the classification of each of the individual indices, it can be assumed that these contamination levels are due to the presence of As and Pb in both lagoons. On the other hand, the correlation between the SRI

of each index evaluated and the SRI of the surface water of the Tragadero lagoon ranged from −0.19 to −0.15 (Figure S1a) and of the Paca lagoon from −0.24 to −0.17 (Figure S1b).

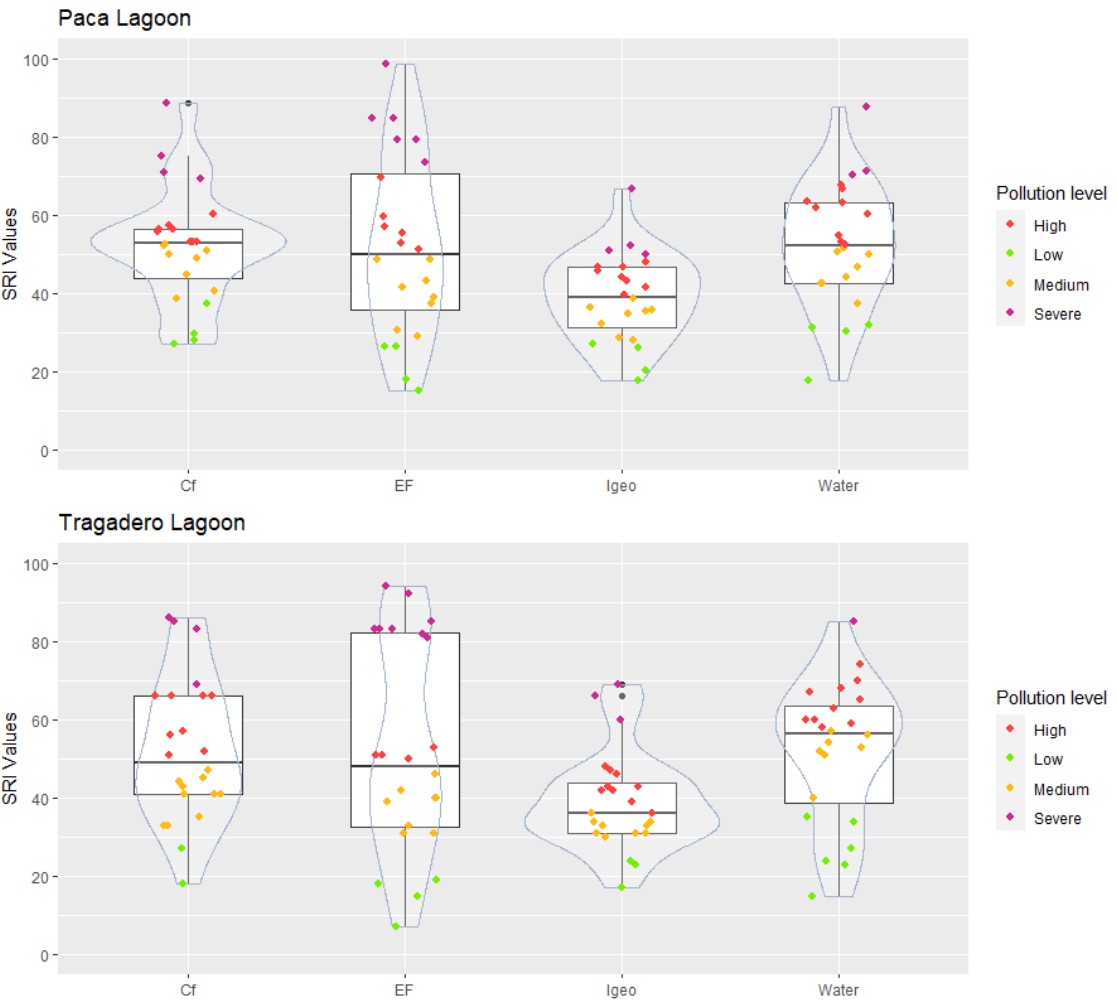

**Figure 3.** SRI distribution to each individual index and the concentration of metals in the surface water.

The complex indexes, pollution load index (PLI), and modified degree of contamination (mCd) allow the information to be condensed from the Cf calculated for each metal evaluated in the sediment samples [26,38]. These indices project a general classification of the level of contamination. However, the mCd categories (seven categories) turned out to be more detailed than those of the PLI (three categories). The calculated PLI values indicated environmental impairment at all sampling sites in both wetlands, whereas the mCd values indicated a degree of contamination ranging from moderate to high. The potential ecological risk (Ri) was calculated to evaluate the degree of sediment contamination by heavy metals and the response of the environment [57]. The ecological risk (Er) values for As were moderate and considerable in the Paca lagoon, and considerable and high ecological risk in the Tragadero lagoon (Table 2 and Figure 4). The Er values for Pb and Zn showed low risk levels. According to the calculated potential ecological risk index (Ri), 79% of the sampling sites in the Paca and Tragadero lagoons showed moderate Ri. As was the toxic element with the highest contribution to Ri, followed by Pb. The analysis of these indices showed contamination in both lagoons, and is related to the level of contamination found in the SRI calculation. However, the mCd categories indicate that the Tragadero lagoon is in a more severe state of contamination than the Paca lagoon. On the other hand, the interpretation of the potential ecological risk is usually more specific to the effect of the

evaluated metals on the ecosystem organisms [22], as it considers the biological toxic factor of each metal [58].

The present results also showed a high degree of correlation (r = 0.99) between the values of mCd and Ri for both lagoons. The PLI and Ri values in the Tragadero lagoon showed no correlation, and in the Paca lagoon a medium correlation (r = 0.56) was visualized. Finally, the PLI and mCd values showed a low positive correlation (r = 0.43) for Tragadero Lagoon and high correlation (r = 0.68) for Paca Lagoon (Figure S2). In order to evaluate the behavior of these variables and determine which of the two (PLI or mCd) best complemented what was expressed by Ri, linear regression between the variables was evaluated.

The linear regression analysis for the PLI and Ri variables in both lagoons, detailed in Figure S3, shows that the proportion of the variance "explained" by the regression model expressed in the coefficient of determination ($R^2$) is low, and there is no significant association that would make the predictive model useful from the Ri variable, as the $R^2$ values are low ($R^2 = 0.28$ for the Paca lagoon and $R^2 = 0.07$ for the Tragadero lagoon), which would show complete independence and random distribution between the variables (Figure 5). However, between the values of mCd and Ri, there is a linear relationship between both variables with high regression coefficients ($R^2 = 0.97$ for the Paca lagoon and $R^2 = 0.99$ for the Tragadero lagoon; Figure 4). These results explain that more than 95% of the variation of the potential ecological risk is influenced by the variability of the degree of contamination in both wetlands, making them dependent as the Ri values are linearly associated with the mCd values, allowing for an optimal model and a reliable prediction of the contamination level.

**Table 2.** Ecological risk and potential ecological risk values of the sediment samples collected in the Tragadero lagoon.

| Sample | Tragadero Lagoon | | | | | | Paca Lagoon | | | | | |
| | PLI | mCd | Ecological Risk | | | Potential Ecological Risk (Ri) | PLI | mCd | Ecological Risk | | | Potential Ecological Risk (Ri) |
| | | | Pb | Zn | As | | | | Pb | Zn | As | |
| 1 | 2.21 | 4.52 | 14.88 | 1.01 | 134.84 | 150.73 | 1.70 | 2.92 | 13.49 | 1.49 | 71.9 | 86.87 |
| 2 | 2.4 | 5.27 | 13.86 | 1.19 | 165.26 | 180.31 | 1.65 | 3.10 | 10.55 | 1.48 | 85.55 | 97.57 |
| 3 | 2.42 | 5.63 | 13.36 | 1.16 | 180.59 | 195.12 | 1.57 | 2.73 | 10.55 | 1.45 | 70.63 | 82.63 |
| 4 | 2.18 | 4.08 | 14.47 | 1.14 | 116.93 | 132.53 | 1.71 | 3.44 | 11.47 | 1.44 | 97.5 | 110.41 |
| 5 | 2.07 | 4.07 | 13.87 | 1.11 | 118.83 | 133.82 | 1.65 | 3.47 | 11.06 | 1.33 | 100.91 | 113.3 |
| 6 | 2.21 | 5.22 | 13.09 | 0.92 | 167.59 | 181.6 | 1.83 | 3.85 | 12.58 | 1.33 | 112.39 | 126.3 |
| 7 | 2.18 | 4.39 | 13.62 | 1.04 | 131.77 | 146.43 | 1.74 | 3.55 | 13.05 | 1.32 | 100.17 | 114.54 |
| 8 | 1.89 | 4.13 | 13.33 | 1.08 | 124.15 | 138.56 | 1.75 | 3.49 | 13.49 | 1.31 | 96.83 | 111.62 |
| 9 | 2.03 | 5.14 | 13.45 | 1.16 | 163.83 | 178.44 | 1.63 | 3.43 | 11.68 | 1.31 | 98.28 | 111.27 |
| 10 | 2.39 | 5.91 | 14.05 | 1.02 | 192.19 | 207.26 | 1.66 | 3.50 | 11.53 | 1.3 | 101.47 | 114.3 |
| 11 | 2.28 | 5.54 | 13.36 | 0.96 | 179.26 | 193.58 | 1.54 | 2.73 | 11.48 | 1.19 | 71.61 | 84.27 |
| 12 | 1.93 | 3.68 | 11.92 | 0.94 | 108.07 | 120.93 | 1.53 | 3.00 | 11.3 | 1.16 | 83.23 | 95.69 |
| 13 | 1.98 | 4.23 | 11.16 | 1.09 | 131.03 | 143.28 | 1.54 | 3.04 | 11.31 | 1.12 | 85.09 | 97.52 |
| 14 | 1.81 | 4.96 | 10.5 | 0.96 | 164.52 | 175.98 | 1.52 | 2.71 | 11.75 | 1.12 | 70.88 | 83.75 |
| 15 | 2.08 | 4.71 | 12.14 | 0.89 | 149.32 | 162.35 | 1.59 | 3.71 | 8.89 | 1.11 | 116.89 | 126.89 |
| 16 | 1.87 | 4.97 | 11.37 | 1.04 | 162.49 | 174.9 | 1.62 | 3.35 | 10.59 | 1.1 | 98.83 | 110.52 |
| 17 | 2.07 | 4.99 | 12 | 0.95 | 161.24 | 174.19 | 1.49 | 3.17 | 11.05 | 1.1 | 91.35 | 103.5 |
| 18 | 1.79 | 4.6 | 10.88 | 0.98 | 149.3 | 161.16 | 1.45 | 2.62 | 11.22 | 1.07 | 68.97 | 81.26 |
| 19 | 1.72 | 4.52 | 10.38 | 0.93 | 147.75 | 159.06 | 1.58 | 3.22 | 11.99 | 1.07 | 91.32 | 104.37 |
| 20 | 2.19 | 5.68 | 11.53 | 0.92 | 189.17 | 201.62 | 1.55 | 2.98 | 10.45 | 1.07 | 84.5 | 96.01 |
| 21 | 1.9 | 5.14 | 12.24 | 0.97 | 168.13 | 181.34 | 1.59 | 3.21 | 11.66 | 1.05 | 91.75 | 104.46 |
| 22 | 2.04 | 4.32 | 11.41 | 0.9 | 134.83 | 147.14 | 1.59 | 3.38 | 11.95 | 0.97 | 98.78 | 111.69 |
| 23 | 1.88 | 5.57 | 10.8 | 0.96 | 188.49 | 200.25 | 1.50 | 3.31 | 10.47 | 0.97 | 99.48 | 110.92 |
| 24 | 1.99 | 4.6 | 10.59 | 0.88 | 148.45 | 159.91 | 1.60 | 3.35 | 11.24 | 0.95 | 98.77 | 110.97 |

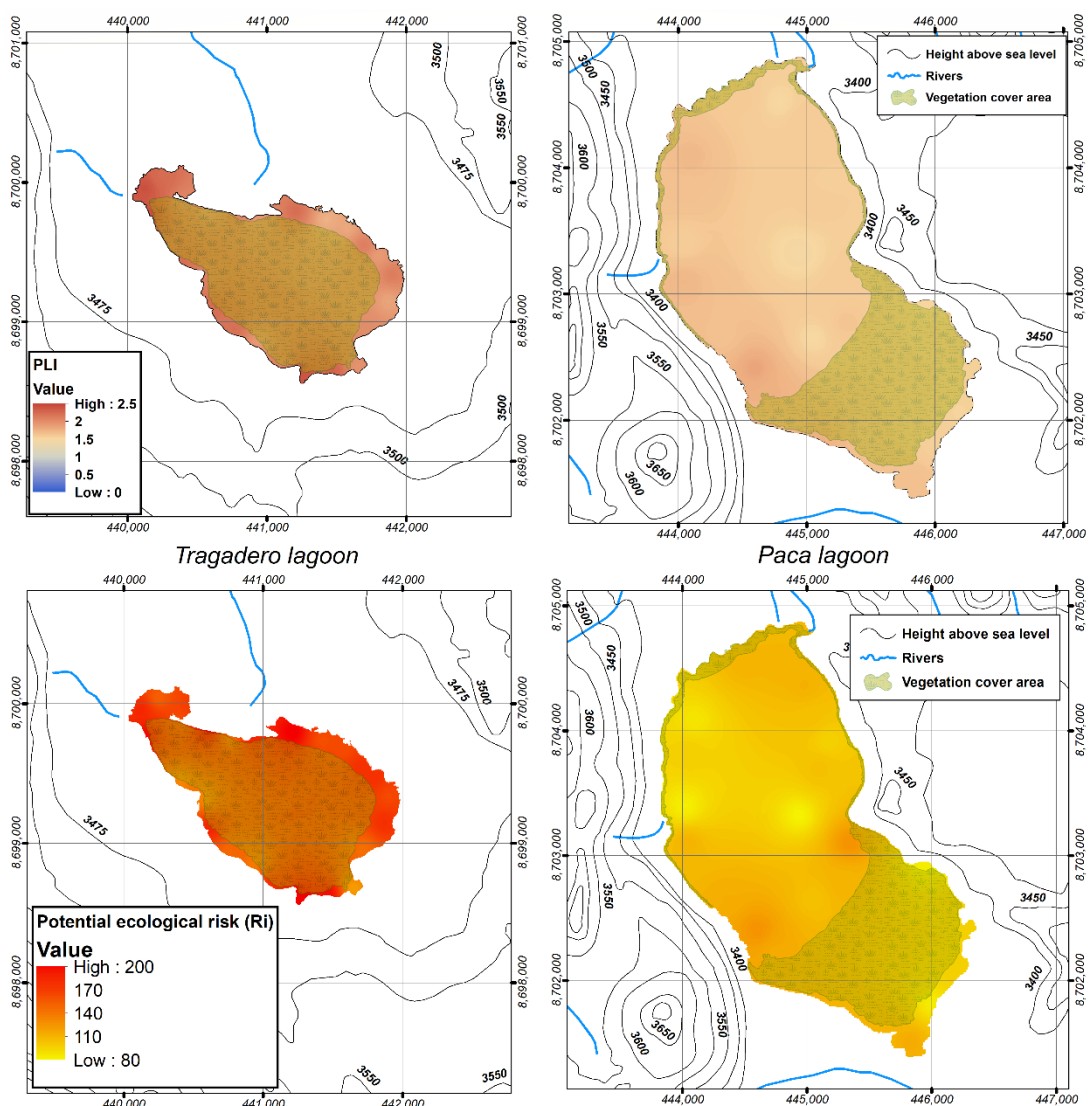

**Figure 4.** Contamination load index (PLI) and potential ecological risk (Ri) of heavy metals in sediments of the Tragadero and Paca lagoons.

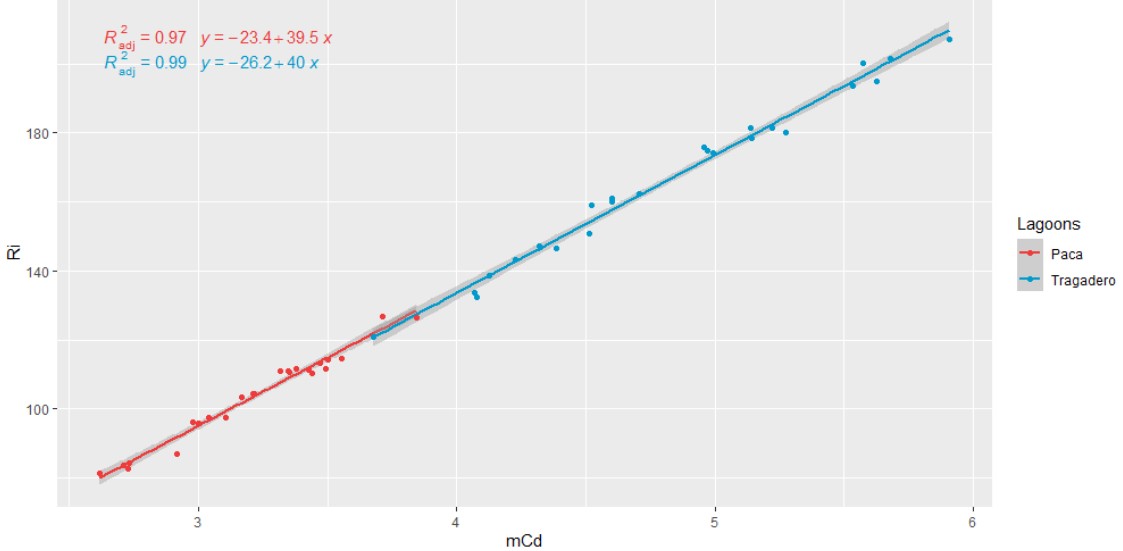

**Figure 5.** Linear regression between modified degree of contamination (mCd) and potential ecological risk (Ri) values of the lagoons.

### 3.3. Identification of Potential Pollutant FigureSources through Principal Component Analysis (PCA)

Principal component analysis (PCA) was performed using the toxic metals that would be generating considerable levels of contamination (As, Pb, and Zn) in the two lagoons. The data were analyzed through the KMO sample adequacy measure (0.55) and Bartlett's test of sphericity ($p$-value = 0.01). Components 1 and 2 covered 79.2% of the total variance (PC1: 49.8% and PC2: 29.4%; Figure 6). From the PCA, a strong association between As, As-W, Pb, and Pb-W concentrations was observed in the Tragadero Lagoon sampling sites, while the Paca sampling sites were better associated with Zn and Zn-W concentrations. On the other hand, the concentrations of As, As-W, Pb, and Pb-W showed a high level of relationship, which could indicate that there is a constant input of As and Pb in both lagoons, generating the high levels of contamination recorded for these metals. In general, the PCA revealed that there are significant loads of As, Pb, and Zn that influence the quality of the sediments and water of the Paca and Tragadero lagoons, mainly from anthropogenic sources (urban discharges and agricultural runoff).

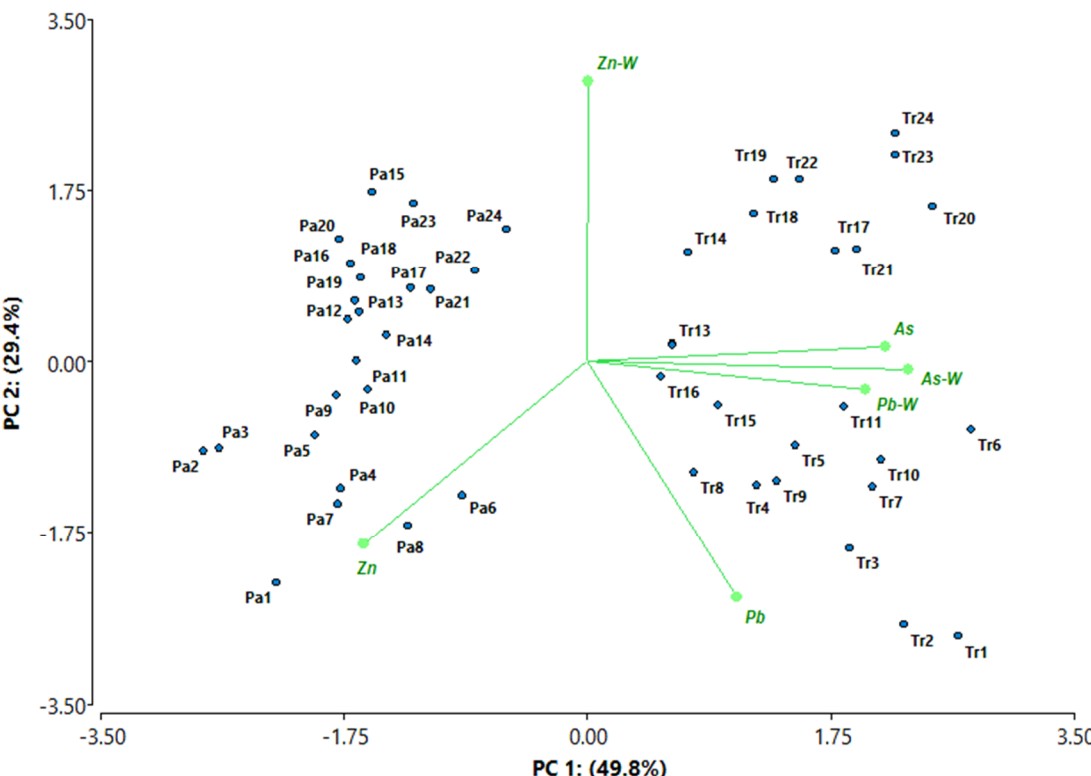

**Figure 6.** Principal component analysis based on the concentrations of toxic metals in the sediments and water samples of the Paca Lagoon (Pa) and Tragadero Lagoon (Tr). Metals with "-W" represent the concentration of metal in the water surface samples.

### 4. Conclusions

The average concentrations of metals in the sediments of the Paca and Tragadero lagoons decreased in the order Fe > Zn > Pb > As. As and Pb concentrations in the sediments of both lagoons exceeded their corresponding background values, while Pb concentrations in the surface water exceeded the levels of environmental water quality standards. Pollution index analyses identified As and Pb as the elements with the highest contribution to environmental degradation (PLI > 1) of the ecosystems by anthropogenic actions (EF > 4). The mCd and Ri values indicate that Paca Lagoon is at a moderate degree of contamination and potential ecological risk, and Tragadero Lagoon is at a high degree of contamination and considerable potential ecological risk. The SRI index for sediments determined a similar distribution of contaminants, with 50% of the sampling sites in

both lagoons being between high and severe contamination. There is a strong correlation between the values of potential ecological risk and the modified degree of contamination. The results of the correlation analysis and PCA identified contributions of As and Pb from anthropogenic sources. Urban activities associated with agricultural activities are responsible for the greatest accumulation of metals in this area. The findings of this study can be used in the adoption of measures to control and reduce the input of metals that generate toxicological effects in the biota of the studied lagoons. The analysis of these data also allows us to reflect on the application and effectiveness of the results obtained through the pollution and potential ecological risk assessment indices for toxic metals.

**Supplementary Materials:** The following are available online at https://www.mdpi.com/article/10.3390/w13162256/s1, Figure S1: Correlation matrix of SRI calculated from each individual index and metal concentration in water (a: Tragadero lagoon; b: Paca lagoon) (***: High correlation degree), Figure S2: Correlation matrix of PLI, mCd and Ri calculated from sediment samples (a: Tragadero lagoon; b: Paca lagoon) (***: High correlation degree, **: Medium correlation degree, *: Low correlation degree), Figure S3: Linear regression between PLI and Ri values of the lagoons, Table S1: Description of individual indices, methods and categorizations used in sediment assessment, Table S2: Description of complex indices, methods and categorizations used in sediment assessment, Table S3: Metal concentration in sediments and surface water of samples collected in Tragadero lagoon, Table S4: Metal concentration in sediments and surface water of samples collected in Paca lagoon.

**Author Contributions:** Conceptualization, investigation, methodology, and writing—original draft, M.C. and A.F.; methodology, F.C. and E.O.-M.; data curation and formal analysis, R.P. and J.C.A.; methodology, D.C. and S.P. All authors have read and agreed to the published version of the manuscript.

**Funding:** This research received no external funding.

**Institutional Review Board Statement:** Not applicable.

**Informed Consent Statement:** Not applicable.

**Data Availability Statement:** Not applicable.

**Acknowledgments:** We thank the Water Research Laboratory for allowing us to use their equipment and materials to carry out the sampling phase.

**Conflicts of Interest:** The authors declare that they have no known competing financial interest or personal relationships that could have appeared to influence the work reported in this paper.

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
