# Peer review of "Ecological Risk Due to Heavy Metal Contamination in Sediment and Water of Natural Wetlands with Tourist Influence in the Central Region of Peru"

_water, doi:10.3390/w13162256_

Round 1
Reviewer 1 Report
Comments:
In this manuscript, the quality of sediment and surface in the wetlands was studied using several index of heavy metals (Fe, Zn, Pb, As). The relationship with the concentration of heavy metals in water was determined, and the sources of contamination by the combination of multivariate statistical analysis were found. In general, the results are well presented and the conclusion has useful implication for understanding the contamination of metals hetals in the water environment ecosystem. Thus, this paper could be published. However, minor revision was needed before this manuscript could be accepted.
- At the end of Line 83, the punctuation is missed.
- Line 120, “0.45 mm membrane filters” should be revised “0.45 μm membrane filters”.
- The result needs to be condensed into three points.
- Please edit the references section more carefully, to make sure that all references are formatted consistently as required by the journal, that spelling is correct, and that the dates for each reference are the same in the text and in the references.
Author Response
First of all, the authors appreciate the reviewer's time and effort in reviewing our manuscript. We are pleased to know that the reviewer considers that the results of our manuscript are well presented and the conclusion has useful implications for understanding metallic element contamination in the aquatic environment ecosystem. The authors appreciate the reviewer's comments and are grateful for helping to refine the manuscript. The authors have responded to each of the comments and have made appropriate revisions to the manuscript.

Reviewer 2 Report
The paper is very well organized. It presents very interesting results of trace element concentrations in bottom sediments and water samples in two natural wetlands Paca and Tragadero located in the central region of Peru. The tables and figures are very well prepared and properly described.
Main comments:
- Line 35-43. Text should be deleted „The introduction should briefly place the study in a broad context and highlight why it is important. It should define the purpose of the work and its significance. The current state of the research field should be carefully reviewed and key publications cited. Please highlight controversial and diverging hypotheses when necessary. Finally, briefly mention the main aim of the work and highlight the principal conclusions. As far as possible, please keep the introduction comprehensible to scientists outside your particular field of research. References should be numbered in order of appearance and indicated by a numeral or numerals in square brackets—e.g., [1] or [2,3], or [4–6]. See the end of the document for further details on references”.
- Line 66-68. Prepare in accordance with the guidelines for authors.
- Line 88. Change o to o.
- Description of symbols from formulas 1 to 7 in the text should be improved (italic).
- Line 91. Change to italic Scirpus californicus.
- Line 139, 147, 149 The unit notation should be improved.
- Complete the labels of the measurement points in Figure 1.
- The description of the measurement points in Figure 2 should correspond to the description of the measurement points in Figure 1.
- Table 2 description of the measurement points should correspond to the description in Figure 2.
- Line 363, 385 and 386 The description of R2 should be improved.
- Complete the data for Paca lagoon in Table 2.
- The paper should be completed with a spatial presentation of the results against the background of the locations of the measurement points.
- In Figure 1 it is useful to mark potential sources of pollution, mark the hydrographic network and the direction of river flow.
- Complete the description of hydrologic conditions regarding river discharge into the Paca and Tragadero wetlands.
- Complete the description of bottom sediment texture (organic matter, sand, clay and silt). How trace element concentrations change in the context of bottom sediment texture?
Author Response
First of all, the authors appreciate the reviewer's time and effort in reviewing our manuscript. The authors appreciate the reviewer's comments and are grateful for helping to refine the manuscript. The authors have responded to each of the comments and have made appropriate revisions to the manuscript.
